# Health shocks, medical insurance and household vulnerability: Evidence from South Africa

**Pheeha Morudu** [ID][◎], **Umakrishnan Kollamparambil** [ID]*[◎]

School of Economics and Finance, University of the Witwatersrand, Johannesburg, South Africa

◎ These authors contributed equally to this work.
* uma.kollamparambil@wits.ac.za

**Data Availability Statement:** The data underlying the results presented in the study are available from http://www.nids.uct.ac.za/nids-data/data-access.

## Abstract

### Background

South Africa has a dual system of healthcare model differentiated across socio-economic lines. While on the one hand there exists high quality private facilities that is expensive and accessible to the minority, on the other is the free but stretched and over-crowded public healthcare that the rest of the population relies on. Accessing private facilities requires private medical insurance or requires coping strategies that can lead to household vulnerability.

### Objective

The objective of this study is to analyse the relationship between health shocks and household vulnerability in the South African context of high poverty and low medical insurance penetration rate.

### Data

The study employs data from waves three to five of South Africa's nationally representative National Income Dynamics Study (NIDS) conducted between the period 2012–2017 in approximately two-year intervals.

### Methods

Using food expenditure shock as an indicator for vulnerability, the study utilises a range of econometric techniques from panel logit regression to quasi-experimental design based difference in difference regressions to assess the association between health shocks, medical insurance and household vulnerability.

### Findings

The main finding of the study is that a significant proportion of households in the upper income quintile utilise private healthcare even when not covered by private medical insurance. This preference for private over public health facilities make them vulnerable to health

**Funding:** The authors received no specific funding for this work.

**Competing interests:** The authors have declared that no competing interests exist.

shocks as they cope by sacrificing food consumption to incur additional health expenditure. In contrast, lower income households that are unable to access the high-cost private healthcare tend to rely on the strained public healthcare system. They are not able to use their constrained food expenditure as a coping strategy for private or out-of-pocket medical expenses because their food consumption is already at a bare minimum.

## Conclusion

The results confirm that access to quality healthcare is a privilege in South Africa, available only to the minority of the population. The study paints a grim picture of household vulnerability in South Africa and underlines the need for a National Health Insurance that would enable universal access to quality healthcare in the country.

## Introduction

The third goal of the United Nations' 2030 agenda for sustainable development aims at ensuring that all people have access to quality health services, while goals one and two are aimed at eliminating poverty and hunger respectively. These three goals are interlinked as poor and uninsured households are vulnerable to health shocks which pushes them deeper into poverty [1, 2]. The twofold mechanisms of the increased burden of out-of-pocket health spending [3, 4]; and decreased labour supply in terms of both hours and labour participation rate link health shocks to household vulnerability [5, 6], forcing households to adopt coping mechanisms like reducing non-medical, non-food consumption as well as food consumption [7–11]. This implies that insurance, which is intended to cover individuals against health shocks, is important in preventing extreme economic outcomes [12].

South African healthcare system currently has a dual model of high-quality private facilities that is accessible to a minority and the stretched, over-crowded public facilities that the majority of population have to rely on. Public healthcare facilities are free to all residents, but residents have the option to purchase private insurance in order to be treated at private facilities. The Parliamentary Monitoring Group [13] reported after an inspection of public health facilities that "Common to all facilities were challenges regarding patient safety being compromised, good pharmacy practice not being adhered to, waste mismanagement, lack of cleanliness, as well as poor maintenance of grounds and equipment". Other studies comparing public and private facilities [14, 15] highlight the major gap between private and public healthcare in South Africa. Public healthcare has many disadvantages such as long wait times, poor quality of care, rushed appointments, old facilities, and poor disease control and prevention practices [14]. Private healthcare on the other hand is expensive [14, 15], but provide quality facilities and care. Peltzer & Phaswana-Mafuya [14] indicates that healthcare responsiveness perception was higher in private than in public inpatient and outpatient healthcare facilities among older South Africans.

The General Household Survey (GHS) reported that only 16.9 percent of individuals in South Africa were covered by private medical insurance in 2017 [16]. This, together with the persistent headcount poverty rates of above 50 percent [17], implies that a large proportion of South Africans have to rely on the inadequate public facilities or have to put in place coping strategies to access private healthcare making them vulnerable to a sudden health shock. This solidifies the need for government intervention in the provision of national health insurance

in the country [17]. South Africa remains in the planning phase of a National Health Insurance (NHI) that will ensure that all South Africans have access to health services [18]. However, there remains several delays in rolling out the proposed NHI scheme, especially with garnering sufficient public funding [19]. In addition, the successful implementation of the NHI requires a transformation to legislation that will enable the establishment of NHI Fund which will be used to fund and sustain the scheme [19].

Against this backdrop, understanding the association between health shocks, household vulnerability and medical insurance will provide context and urgency to the NHI discourse in South Africa. This study, which is the first of its kind in South Africa, contributes to the global literature on the household effects of health shocks by using methodological innovations in terms of defining vulnerability risk and estimation strategies within a quasi-experimental difference-in-difference framework. Studies undertaken in different country contexts show the negative effects of health shocks on income, food expenditure and non-food expenditure [9, 20–22]. Contrary to this, some studies [23, 24] indicate increased food expenditure due to preferences for special food to assist with recovery from illness. Further, the two-way causal relationship between food and health often arising from medical poverty trap also needs to be emphasised. A medical poverty trap refers to the self-perpetuating state where the poor are at greater risk of ill health and the ill health in turn increases the likelihood of becoming poor through out of pocket costs for public and private health services [25–27]. Given the two-way causal relationship, it needs to be emphasised that this study is focussed on understanding the association between health shock and food expenditure shock, and does not make claims of causal effect.

The studies cited above assess the association between health shock and food expenditure but none of them consider the relationship between health shock and food expenditure shock. This study introduces a new measurement variable of vulnerability with respect to food expenditure shocks. Food consumption per capita is considered to be a preferred measure of absolute poverty for developing countries [28] also evidenced by Statistics South Africa which includes a measure of poverty using the food poverty per person [17]. Therefore, this research focuses on significant (more than one standard deviation) declines in food expenditure per person as a food expenditure shock, which represents household vulnerability. This food expenditure shock is a binary variable indicating whether the household experienced a significant decrease in per capita real food expenditure. The study uses three consecutive waves of the National Income Dynamics Study (NIDS) panel data to investigate this relationship. The investigation utilizes waves three to five, conducted in approximately two year intervals between 2012 and 2017. The independent variable (health shock) used in this study is the significant (more than one standard deviation) decline in body mass index (BMI) of non-obese individuals.

## Analytical framework

Existing literature has used different combinations of variables as measures of health shocks such as; a) death of a working age adult household member due to illness [11, 21, 22, 29, 30]; b) self-reported serious illnesses that prevented a household member from being able to work [5, 8, 11, 31] and c) ability to perform daily living activities [7, 32].

However, there are limitations to the NIDS data regarding the above suggested health shock measures. Serious illness as a health shock is defined as any illness that prevented a household member from doing normal activities, serious illness can include any health problems, for example, disability, disease, injury or any other chronic diseases [5, 11, 33]. The NIDS questionnaire changed from wave 4 and the question that asks whether there was a "serious illness or injury of a household member" was discontinued.

The study therefore uses a significant decrease in body mass index (BMI) of non-obese household heads as a health shock measure following Wagstaff [20]. James et al. [34] states that BMI is a recognized and reliable measure of the current nutritional status of adults. Rather than using low BMI scores (representing underweight individuals, implying they were ill from the outset), the health shock variable is defined as a binary variable taking the value one if the BMI score reduced by more than one standard deviation in each period, and zero otherwise. By excluding the obese individuals, the study accounts for the high proportion of obesity in South Africa resulting from very factors including diabetes and also unhealthy diet [35]. The association of both diabetes and food habits with socio-economic status is debatable [36, 37]. A reduction in BMI of obese individuals is not undesirable and hence the study chooses to exclude it from the definition of health shock.

Similar to health shocks, there are different possible measurements for household vulnerability. Hoddinott et al. [38] define vulnerability as the likelihood that an individual will have a level of welfare below a given benchmark at a given time in the future. They further discuss three methods which are used to estimate vulnerability, a) Vulnerability as Low Expected Utility (VEU), b) Vulnerability as Expected Poverty (VEP) and, c) Vulnerability as Uninsured Exposure to Risk (VER) [38]. Ligon et al. [39] define VEU as the difference between the utility derived from a certain level of certainty-equivalent consumption at and above which the household would not be considered vulnerable. Chaudhuri et al. [40] define VEP as the probability that expected consumption expenditure of a household will fall into poverty in the future. Hoddinott et al. and Ligon et al. [38, 39] define VER as a method that assesses welfare loss in the absence of effective risk management tools. The VER approach allows an *ex-post* evaluation of the scope of the negative shock causing a loss of well-being using panel data. The current study is based on the VER measure of vulnerability, adapted from the general VER model [38] in Eq 1 as follows:

$$\Delta \ln C_{htv} = \sum_i \lambda_i S(i)_{tv} + \delta X_{hv} + \Delta \varepsilon_{hvt} \tag{1}$$

Which denotes that a change in log consumption per capita of household $h$, in period $t$ is a function of health shocks $S(i)_{tv}$, and $X_{hv}$, a vector of household or household head characteristics. This also means that when the coefficient $\lambda$ is equal to zero, it implies immunity from health shocks.

## Methods

Broadly, two approaches—a three period panel data and a quasi-experimental data design—are used for analysis. The quasi-experimental data design based difference-in-difference method which uses a non-randomised assignment to treatment or control group in order to estimate the impact of drop in BMI to the sample. In this study, a quasi-experimental sample was used whereby outcomes between a treatment and control groups were compared for two pre-treatments and one post-treatment periods. Model I (Panel Logit)

The following method will use the first dataset, which allows the health shock to be measured as a dummy variable in the three investigation periods based on changes in BMI from the previous period. The panel logit model using binary dependent variable framework is as follows:

$$y_{it} = \beta_0 + \beta_1 (health\ shock_{it}) + \beta_3 X_{it} + \varepsilon_{it} \tag{2}$$

Where:

- Individual and time identifiers are subscripted as $i$ and $t$ respectively,

- $y_{it}$ represents the food expenditure shock dummy variable taking value 1 if there was a significant decrease in food expenditure and 0 otherwise,

- $X_{it}$ is a vector of control variables and,

- $\varepsilon_{it}$ is a non-zero residual variable containing both conditional errors and time invariant unobservables

Due to the absence of an appropriate counterfactual, the estimator is limited, which may bias results in the presence of self-selection [41]. In addition, given the panel data structure and potential two-way causal relationship between food shocks and health shocks, there is the likelihood that control variables and the treatment variable are correlated with time invariant unobservables captured in the composite error. Some of these limitations are accounted for in a difference-in-difference (DID) estimation. The first difference which is usually called the 'naïve' estimator can either compare the treated individuals pre and post treatment or compare treated and untreated individuals post treatment. The problem with differencing only once is that the estimator yield biased estimate of the treatment and differencing does not account for the limitations mentioned initially. Therefore, application of double differencing estimator compares treatment and control groups in terms of outcome changes over time relative to the outcomes observed for a pre-intervention baseline in order to isolate the treatment effect [42].

## 1.1. Model II (DID)

Using the quasi-experimental data design (where waves 3 and 4 are pre-treatment, and wave 5 is the post-treatment period), the difference-in-difference (DID) model can be used to estimate the effect that the health shock has on the treated (i.e. households experiencing health shocks) in the post-treatment period. DID is usually applied to continuous dependent variable, however, in this case, the dependent variable is a binary variable (with value 1 for households that experienced food expenditure shock and, 0 for the rest of the households). Therefore, a non-linear DID model is applicable in this study. Karaca-Mandic et al. [43] suggest that a non-linear DID model should have the conditional probability that the dependent variable equal to 1 be expressed as a function of the usual DID function with continuous dependent variable. The non-linear DID model is represented as follows:

$$P(y = 1|x) = F(\beta_0 + \beta_1 Post + \beta_2 Treat + \beta_{12}(Treat * Post) + X\beta) \tag{3}$$

While for a linear DID a positive and significant $\beta_{12}$ indicates the vulnerability of households to health shocks, the treatment effect in non-linear DID is the difference between the cross differences for each outcome category [44]. Hence the probabilities associated with the marginal cross-difference, used to ascertain the treatment effect, are presented along with the DID results.

The DID estimation is based on the parallel trend assumption which is tested following Autor's [45] methodology. The results presented in S2 Table in the supplementary information, shows insignificant interactions in period 1 implying that the assumption of the parallel trend is satisfied in our dataset. Given that the assignment mechanism between the treatment and control groups is not random, the estimator suffers from the potential for self-selection based on unobservables [45]. This necessitates that measures be taken to account for potential structural differences between the two cohorts. However, there exists a dimensionality problem, that is; it is not clear how each characteristic should be weighted when matching characteristics between treatment group and control group. Rosenbaum and Rubin [46] suggest the use of propensity score matching (PSM) to solve the problem of multi-dimensionality which is undertaken in Model III.

## 1.2. Model III (PSM-DID)

The PSM technique involves a construction of an artificial control group by identifying an untreated observation that has the most similar observable characteristic for every treated observation. A stepwise logistic regression to select variables was estimated, as proposed by Rosenbaum and Rubin [46]. This resulted in significantly related variables in the final model. The matching algorithm employed in this study uses the nearest neighbour matching algorithm which is also known as 'traditional pairwise matching'. The common support, or overlap condition, was tested and found to be satisfactory (Fig 1).

The conditional independence assumption (CIA) assumes that the spread of the propensity scores is identical for those receiving treatment and not receiving treatment because all factors that generate dependence is the same. However, the CIA is always assumed in most cases because it cannot be tested in practice since it involves unobserved potential outcomes. S3 Table shows that the entire samples' matched propensity scores result in a reduction in selection bias.

Despite attempts to minimise endogeneity, the concern of reverse causality remains. Added to this, due to data limitations, using an instrumental variable approach to reducing endogeneity was difficult to implement. Thus the results of the empirical analysis are best interpreted as association rather than causation.

## Data

The data used in the investigation is extracted from the National Income Dynamics Study (NIDS) of South Africa. Supervised by the Southern Africa Labour Development Research

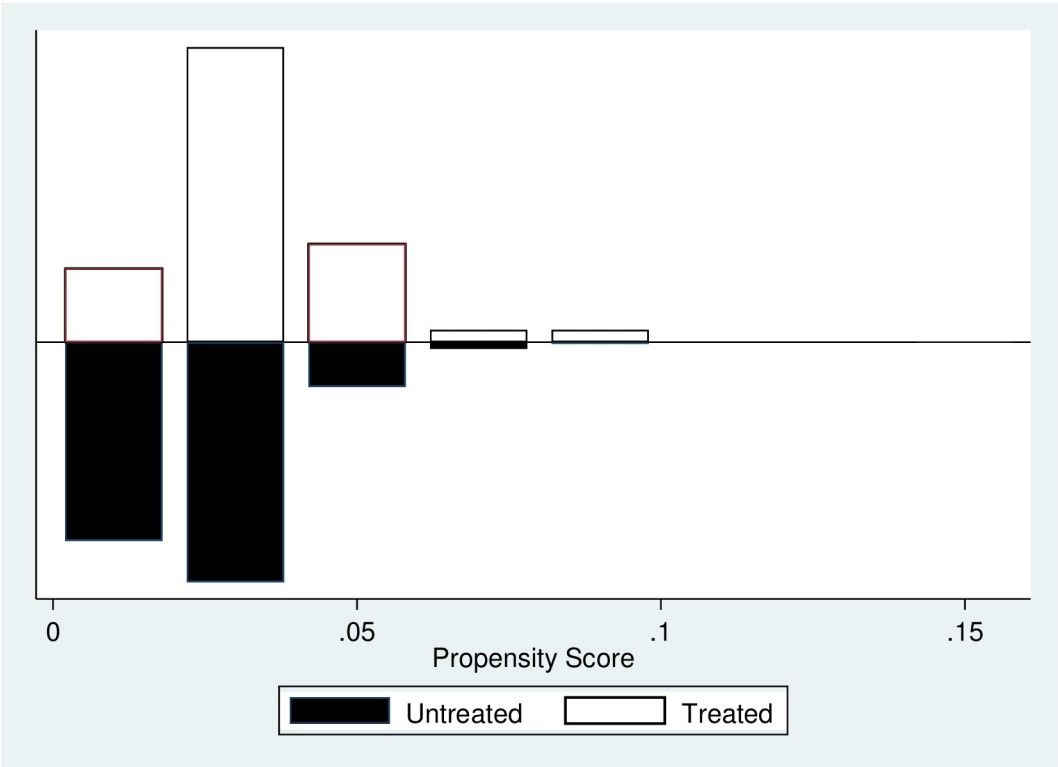

**Fig 1. Common Support for entire sample.** The final propensity score matching drops the observation that are off-support. The propensity scores were obtained from Stata command *psmatch2* which were utilised in the estimation of the following PSM-DID model. However, it is worth noting that PSM reduces but does not eliminate the bias generated by unobservable confounding factors.

Unit (SALDRU) [47], the NIDS dataset is a national representative panel study and currently has five waves with approximately 2 year intervals. The panel dataset has approximately 7 300 households and 28 000 individuals captured in the database. To ensure continued national representation of the population, the NIDS dataset contains panel weights to account for systematic attrition and non-responses.

The study uses two structures of data derived from NIDS to suit the different estimation strategies. First, a three period panel dataset of non-obese household heads is constructed using waves 3 to 5. Shocks in health and food expenditures for wave 3 is estimated based on changes from wave 2. Therefore, although only waves 3–5 are included in the analysis, wave 2 is utilised for key variable construction. The second dataset is a quasi-experimental dataset using only waves 3 to 5, where treatment is defined as the non-obese household heads suffering from a health shock between period 4 and 5. The sample is restricted to those not suffering from health shocks in waves 3 and 4. As such, waves 3 and 4 are constructed as pre-treatment periods and wave 5 as the post-treatment period. Detailed definition of variables is provided in S1 Table in the supplementary file.

Households experiencing health and food expenditure shocks calculated in the panel dataset is summarised in Table 1.

The quasi-experimental dataset (used for DID) restricted the sample such that health shocks which represent a significant reduction in the BMI are observed only between the 4th and 5th waves, and divided the sample between treatment and control groups. The treatment group consists of non-obese household heads who experienced an over 2.9-point decline in BMI in wave 5.

Further, various households and household head characteristics are included as control variables. The household characteristics include location, household size and household income; while household head characteristics include age, sex, education, medical aid coverage and employment status. Table 2 contains the average weighted means and standard errors for all variables used in the quasi-experimental dataset. Table 2 illustrates observable differences in variables between treatment and control groups for the three periods. The following has been divided between those who experienced health shocks, the 'Treated' group, and those who did not experience a health shock, the 'Control' group. The dataset contains 4 812 observations.

On average, 2.56 percent of households experienced a health shock in wave 5. There is no significant difference in the proportion of sample that experienced health shocks by their medical insurance status. Evidence from literature is mixed in terms of the effect of medical insurance on health status, with a recent review [48] indicating that of the 12 studies: nine studies found a positive effect, one study reported a negative effect, and two studies reported no effect. Therefore the indication that medical insurance does not make a substantial impact on health shock is not contradictory to existing evidence.

Food shocks captures whether the household experienced a significant decrease in real total food expenditure or not. Food shocks were experienced by 8.20 and 8.80 percentages of treated and control group households respectively. Household income per capita and food

**Table 1. Percentage experiencing health shocks and food expenditure shocks.**

| Variables | Wave 3 | Wave 4 | Wave 5 |
|---|---|---|---|
| Health shock | 1,96 | 3,50 | 2,73 |
| Food Expenditure Shock | 6,89 | 9,67 | 10,70 |
| Number of Households | 1 684 | 1 686 | 1 683 |

Source: Authors' calculation from weighted NIDS data

**Table 2. Summary statistics of DID sample.**

| Variable | | N | | Mean | | Std. Error | |
|---|---|---|---|---|---|---|---|
| | Sample | Treated | Control | Treated | Control | Treated | Control |
| *Household characteristics* | | | | | | | |
| Health Shock | Whole | 4812 | | 0.026 | | 0.157 | |
| Food Shock | | 123 | 4 689 | 0.082 | 0.088 | 0.276 | 0.283 |
| Income | | 123 | 4 689 | 4019,62 | 3927,71 | 4383,36 | 8016,45 |
| Food Expenditure | | 123 | 4 689 | 608,88 | 521,69 | 651,25 | 605,85 |
| Household Size | | 123 | 4 689 | 3.86 | 3.75 | 2,55 | 2.67 |
| Urban | | 123 | 4 689 | 0,5905 | 0,6897 | 0,4938 | 0,4627 |
| *Household head characteristics* | | | | | | | |
| Medical Aid Coverage | | 123 | 4 689 | 0,0764 | 0,1598 | 0,2668 | 0,3664 |
| Female | | 123 | 4 689 | 0,2856 | 0,5239 | 0,4535 | 0,4995 |
| Married | | 123 | 4 689 | 0,3794 | 0,1393 | 0,4872 | 0,3463 |
| Age | | 123 | 4689 | 54.64 | 50.08 | 14.2 | 14.9 |
| Education | | 123 | 4 689 | 6.6 | 8.48 | 4.49 | 58050 |
| Pensioner | | 123 | 4 689 | 0,2257 | 0,1367 | 0,4197 | 0,3436 |
| Employed | | 123 | 4 689 | 0,4238 | 0,5560 | 0,4962 | 0,4969 |

Source: Authors' calculation from weighted NIDS data

expenditure per capita have large deviations and is indicative of the high level of inequality in the country. From above, on average only 7.64 percent of those belonging to the treatment group are covered by a medical aid while 16 percent of those in the control group are covered by medical aid. The average age of the household head is 50 years old. The low average levels of education attachment, especially among the treated group, is also evident. Similarly, lower percentage of household heads are employed among the treated as compared to the control group. This shows that health shocks can significantly reduce labour supply; as highlighted by [5, 8, 49, 50]. This indicates that the mechanism of the health shock transmission to food shocks therefore could be from foregone labour, as well as out-of-pocket medical expenditure.

**Table 3. Summary statistics of household and individual characteristics (average weighted, Quintile).**

| Variable | N | | Mean | | Std. Error | | Min | | Max | |
|---|---|---|---|---|---|---|---|---|---|---|
| Sample | Quintile 1 | Quintile 2 | Quintile 1 | Quintile 2 | Quintile 1 | Quintile 2 | Quintile 1 | Quintile 2 | Quintile 1 | Quintile 2 |
| Total Medical Expenditure | 3 094 | 1 718 | 23,34 | 101,38 | 107,54 | 775,50 | 0,00 | 0,00 | 3067,49 | 10224,95 |
| HH Food Expenditure | 3 094 | 1 718 | 251,18 | 796,38 | 235,85 | 730,09 | 0,00 | 0,00 | 4089,98 | 8179,96 |
| HH Income | 3 094 | 1 718 | 747,19 | 7116,90 | 387,86 | 10307,57 | 51,12 | 1549,03 | 1547,38 | 152777,80 |
| Urban | 3 094 | 1 718 | 0,5740 | 0,8012 | 0,4946 | 0,3992 | 0 | 0 | 1 | 1 |
| Household Size | 3 094 | 1 718 | 4,1252 | 1,8466 | 2,6993 | 1,4807 | 1 | 1 | 24 | 22 |
| Age | 3 094 | 1 718 | 48,3844 | 44,2585 | 13,9490 | 12,6682 | 17 | 17 | 91 | 90 |
| Head Education attainment | 3 094 | 1 717 | 7,6603 | 13,0562 | 4,7068 | 5,5409 | 0 | 0 | 24 | 24 |
| Employed | 3 094 | 1 718 | 0,3336 | 0,7729 | 0,4716 | 0,4191 | 0 | 0 | 1 | 1 |
| Food Expenditure Shock | 3 070 | 1 701 | 0,1116 | 0,0470 | 0,3149 | 0,2116 | 0 | 0 | 1 | 1 |
| Health Shock | 3 094 | 1 718 | 0,0190 | 0,0243 | 0,1365 | 0,1541 | 0 | 0 | 1 | 1 |
| Medical Aid Coverage | 3 094 | 1 718 | 0,0147 | 0,3015 | 0,1204 | 0,4590 | 0 | 0 | 1 | 1 |
| Pension Recipient | 3 094 | 1 718 | 0,2182 | 0,0590 | 0,4131 | 0,2356 | 0 | 0 | 1 | 1 |

Source: Authors' calculation from weighted NIDS data

Table 3 combines the household and individual characteristics per quintile. The total monthly medical expenditure for higher income households on average is approximately four times higher than that of lower income households. Possible explanation for this is that higher income households (even in the absence of medical insurance) prefer private healthcare which are generally more expensive than the public healthcare facilities. This is supported by the statistic that 48% of the quintile 2 individuals that accessed private healthcare, did not have medical insurance. On average, 1.47 percent of lower income household (quintile 1) heads are covered by medical insurance while 30.15 percent of higher income household (quintile 2) heads are covered by medical insurance. One of the limitation of NIDS data however is that details regarding the medical insurance scheme is not provided. We therefore have to treat this as a binary variable.On average, approximately 2 percent of the households experienced a health shock.

The average household size is higher for the quintile 1 (lower per capita household income sample) households compared to the higher income households. The higher income household heads have, on average, higher years of education attainment. The education attainment is a cardinal variable denoting the years of education obtained. Therefore, it is not surprising that, on average, there is a higher percentage of employed heads of households in higher income quintile than those in lower income households. The eligibility for old age pension grant is income below R156 240 per annum for married and R78 120 per annum for single individuals [51]. Thus, it is not surprising that on average, only 6 percent of household heads in higher income quintile while 22 percent of lower income households receive the old age government grant. The average household income per capita of quintile 1 households is (R747.19). In contrast, the higher income households have on average monthly income per capita that is almost 3 times more than monthly old age grant.

## Results

Given the binary nature of the dependent variable (food shock) that measures vulnerability, various non-linear panel data models are estimated for the entire sample and sub-samples based on household per capita income. The parallel trend assumption is satisfied for the use of DID regression (S2 Table). Further the propensity score matching tests satisfy the assumptions for PSM-DID and are presented in S3 Table. Results from the panel logit, DID and PSM-DID are presented in Tables 4, 5 and 6 respectively. The results show remarkable consistency with respect to the strong positive and significant association between health shocks and food shocks among quintile 2 households (higher per capita household income sample) and lack of association between health shocks and food shocks among the quintile 1 (lower per capita household income sample) sample. The results strongly suggest that higher income households respond to health shocks by reallocating household spending towards the sick household member and allocating away from overall household food expenditure [20]. This reallocation of household resources is also visible in higher income households that access private healthcare that involves out-of-pocket expenditure which they try to manage by sacrificing food consumption. This is in line with several studies [9, 11, 22, 23, 52] who also found that the health shocks decrease food consumption by 1.80%, 17.30%, 4.80%, 15.30%, 18.80% and 4.20% respectively. While studies [23, 24] have indicated increased food expenditure incurred for special food for the patient, this study shows a reduction in the overall household per capita food expenditure following a health shock.

Our results contradict a study conducted by Gertler et al. [7], which concluded that households with higher income seem to be better insured against negative effects of illness shocks.

**Table 4. Panel logit model outputs—Model I.**

| | Whole sample | Quintile 1 | Quintile 2 |
|---|---|---|---|
| Health shock | 0.2977 | -0.2141 | 1.1417* |
| | (0.3812) | (0.4983) | (0.6860) |
| Medical Aid coverage | 0.0602 | -1.3793 | 0.5298 |
| | (0.2838) | (0.8456) | (0.3344) |
| Female | -0.0611 | -0.2816 | 0.1554 |
| | (0.1892) | (0.2441) | (0.3102) |
| Married | -0.3558* | -0.2903 | -0.5241 |
| | (0.1968) | (0.2389) | (0.3981) |
| Age | | | |
| 30–49 | -0.0065 | 0.1725 | -0.3172 |
| | (0.2903) | (0.3969) | (0.4387) |
| 50–64 | -0.4262 | -0.3328 | -0.4841 |
| | (0.3260) | (0.4397) | (0.5140) |
| 65–91 | -0.1104 | -0.1137 | 0.5655 |
| | (0.4026) | (0.5306) | (0.6997) |
| Education Attainment | | | |
| Primary | -0.0950 | -0.1574 | -0.0072 |
| | (0.2498) | (0.2819) | (0.6410) |
| Secondary | -0.7806*** | -0.8000** | -1.2086* |
| | (0.2843) | (0.3364) | (0.6558) |
| Certificate | -0.6652* | -0.3515 | -1.2101* |
| | (0.3650) | (0.4753) | (0.7175) |
| Undergraduate | -0.0930 | 0.7380 | -0.5511 |
| | (0.4018) | (0.7427) | (0.7062) |
| Postgraduate | -0.3338 | - | -0.3062 |
| | (0.7391) | | (0.9579) |
| Household size | | | |
| 4–6 | 0.7463*** | 0.7176*** | 0.5237 |
| | (0.1720) | (0.2156) | (0.3562) |
| 7–9 | 0.9801*** | 0.9020*** | 1.1033 |
| | (0.2416) | (0.2811) | (0.7600) |
| 10–24 | 1.2936*** | 1.1886*** | 2.0339 |
| | (0.3625) | (0.4028) | (1.5962) |
| Pension recipient | -0.5299** | -0.4015 | -1.5749** |
| | (0.2567) | (0.3009) | (0.6772) |
| Employed | -0.2855* | -0.4396** | 0.4477 |
| | (0.1697) | (0.2119) | (0.3916) |
| Urban | 0.1444 | 0.1296 | 0.1932 |
| | (0.2398) | (0.3045) | (0.4099) |
| Constant | -3.0914*** | -3.0589*** | -3.8602*** |
| | (0.4507) | (0.5629) | (1.0711) |
| Observations | 5053 | 3248 | 1 801 |

\*\*\* Significant at 1% level

\*\* significant at the 5% level

\* Significant at the 10\* level

**Table 5. Difference in differences model output—Model II.**

|  | Whole sample | Quintile 1 | Quintile 2 |
|---|---|---|---|
| Post | 0.4467** | 0.6489*** | -0.0034 |
|  | (0.1797) | (0.2000) | (0.3598) |
| Treat | -1.5646*** | -1.1333* | -2.7073** |
|  | (0.5522) | (0.5988) | (1.2104) |
| Treat*Post | 2.3725*** | 1.3821 | 5.7218*** |
|  | (0.8638) | (0.9582) | (1.6597) |
| Medical Aid coverage | -0.4324 | -1.8649* | 0.1521 |
|  | (0.3151) | (1.0191) | (0.4207) |
| Female | -0.3529** | -0.6428*** | 0.1123 |
|  | (0.1786) | (0.2009) | (0.3070) |
| Married | 0.1271 | 0.3771 | -0.7280 |
|  | (0.2350) | (0.2590) | (0.5114) |
| Age 30–49 | 0.0535 | 0.1389 | -0.3026 |
|  | (0.3383) | (0.4514) | (0.4852) |
| Age 50–64 | -0.1950 | -0.2339 | -0.1898 |
|  | (0.3848) | (0.4951) | (0.6246) |
| Age 65–91 | 0.4317 | 0.1172 | 1.3573 |
|  | (0.5817) | (0.5855) | (0.9484) |
| Education: Primary | 0.4556* | 0.4894* | 0.1119 |
|  | (0.2557) | (0.2723) | (0.8645) |
| Education: Secondary | -0.1041 | -0.1329 | -0.2871 |
|  | (0.3054) | (0.3476) | (0.7648) |
| Education: Certificate | -0.2325 | 0.1413 | -0.5652 |
|  | (0.4082) | (0.4645) | (0.8959) |
| Education: University | 0.4743 | 1.1142 | 0.3199 |
|  | (0.4234) | (0.7448) | (0.8083) |
| Household size 4–6 | 0.6904*** | 0.3840* | 0.7931* |
|  | (0.1910) | (0.2211) | (0.4239) |
| Household size 7–9 | 0.6020** | 0.2663 | 1.0036 |
|  | (0.2396) | (0.2582) | (0.6124) |
| Household size 10–24 | 0.8836*** | 0.5037 | 2.7845** |
|  | (0.3133) | (0.3395) | (1.1118) |
| Pension recipient | -0.6926* | -0.4501 | -0.8109 |
|  | (0.3762) | (0.3139) | (0.8391) |
| Employed | -0.2983 | -0.2326 | 0.8722* |
|  | (0.2011) | (0.2290) | (0.5026) |
| Urban | 0.1345 | 0.1702 | 0.3577 |
|  | (0.1688) | (0.1922) | (0.3161) |
| Constant | -2.3943*** | -2.0273*** | -3.7639*** |
|  | (0.4143) | (0.5053) | (1.0350) |
| Observations | 4771 | 3067 | 1701 |
| *Cross difference* (P-value) | 0.0693 | 0.2272 | 0.0319 |

*** Significant at 1% level

** significant at the 5% level

* Significant at the 10* level

**Table 6. PSM-DID results—Model III (propensity score weighted).**

| | whole sample | Quintile 1 | Quintile 2 |
|---|---|---|---|
| Post | 0.3262*** | 0.0630 | 1.3604** |
| | (0.1191) | (0.1488) | (0.5706) |
| Treat | -0.2585 | -0.5202 | -13.3631*** |
| | (0.5047) | (0.5430) | (0.6153) |
| Treat*Post | 0.3089 | -1.0599 | 17.1028*** |
| | (0.7156) | (1.1705) | (1.2857) |
| Medical Aid coverage | -0.4790** | -1.0962 | -0.3836 |
| | (0.2175) | (1.0160) | (0.5286) |
| Female | 0.3007** | -0.2629 | 0.8439** |
| | (0.1299) | (0.1643) | (0.4294) |
| Married | -0.0083 | 0.0391 | -0.4356 |
| | (0.1704) | (0.2030) | (0.5216) |
| Age : 30–49 | -0.1317 | -0.1542 | -0.0672 |
| | (0.2079) | (0.3046) | (0.7032) |
| Age : 50–64 | -0.3690 | -0.3635 | -1.3979* |
| | (0.2308) | (0.3333) | (0.7671) |
| Age : 65–91 | -0.3405 | -0.1187 | -4.1163*** |
| | (0.2848) | (0.3960) | (1.2383) |
| Education: Primary | 0.1152 | 0.1670 | -0.5827 |
| | (0.1718) | (0.1884) | (0.8306) |
| Education: Secondary | -0.3041 | -0.0475 | -1.1580 |
| | (0.1972) | (0.2271) | (0.9666) |
| Education: Certificate | -0.1611 | 0.0700 | -2.2624** |
| | (0.2649) | (0.3524) | (0.8786) |
| Education: University | 0.2542 | 0.2708 | -0.7904 |
| | (0.2862) | (0.8044) | (0.8221) |
| Household size 4–6 | 0.7147*** | 0.5060*** | 0.4346 |
| | (0.1319) | (0.1661) | (0.4997) |
| Household size 7–9 | 0.7716*** | 0.6369*** | -1.0352 |
| | (0.1777) | (0.2009) | (1.0420) |
| Household size 10–24 | 0.6637** | 0.4996* | 3.1548** |
| | (0.2637) | (0.2939) | (1.3187) |
| Pension recipient | -0.2098 | -0.2105 | 1.0420 |
| | (0.1895) | (0.2304) | (1.5078) |
| Employed | -0.3673*** | -0.2649 | 0.3902 |
| | (0.1412) | (0.1716) | (0.9611) |
| Urban | 0.1581 | 0.2168 | 0.2089 |
| | (0.1240) | (0.1440) | (0.5505) |
| Constant | -2.0684*** | -2.1681*** | -2.1403 |
| | (0.2761) | (0.3803) | (1.5954) |
| Observations | 4,771 | 2,678 | 1,122 |
| *Cross differences (P-value)* | 0.6681 | 0.3142 | 0.0000 |

*** Significant at 1% level

** significant at the 5% level

* Significant at the 10% level

Our study shows that even relatively higher income households are vulnerable to health shocks given their preference to access better quality private sector healthcare.

The absence of association between health shocks and food shocks among the poorer households shows that they do not have room to further bring down food expenditure to access healthcare that requires out-of-pocket expenditure. Based on this explanation, it appears that the poor forego quality healthcare, as they cannot trade-off their already constrained food consumption further for medical expenses. Literature has also provided various other reasoning for this. Islam & Maitra [53] highlight the role access to microcredit, as a means for consumption smoothing in the presence of health shocks. Mohanan [54] argue that society, particularly from poorer subpopulations, usually intervene when the ability of a household's food consumption smoothing is limited by a shock. The community intervention can be in the form of neighbours and relatives of the household. This is further highlighted by Udry [55], Townsend [56] and Morduch [57] who also confirmed that informal coping mechanisms (borrowing from landowners, relatives' transfers or other support network) are very popular in developing countries because of the absence or malfunctions of formal credit and insurance markets. In the South African context both sets of reasoning are likely to hold. Poorer households access healthcare only in the free public sector which is severely constrained. The NIDS data at hand substantiates this with 84% of lower income household dependent on public healthcare compared to under 44 percent of quintile 2 households. Other non-medical expenses related to accessing healthcare like transport and foregone labour may be compensated by social intervention, but studies on this is limited in the South African context.

The negative association between medical insurance and household vulnerability is brought out at 5 percent significance level for the whole sample. Furthermore, the results show that larger sized households and households with female heads were more vulnerable to food shocks. These results are in line with the other studies that conclude that increase in household size tends to increase vulnerability to poverty in sub-Saharan countries [2, 58]. Education and employment, on the other hand, are seen to be strong protection against household vulnerability on lines of Dhanara [59]. This is consistent with the findings based on Nigeria data that concluded that in general households with heads who have no schooling are vulnerable [58].

## Discussion and conclusion

This study investigated the relationship between household heads' health shocks and household vulnerability as measured by food expenditure shock. Following Wagstaff [22] the study defined health shock as a decline in BMI of more than one standard deviation among the non-obese. The use of BMI as a measure of health shock has limitations as BMI can decrease due to other measures for example, ageing, change in lifestyle, seasonality. The study estimated Panel Logit models, DID models and, PSM with DID models to address selection bias issues.

The results of the three models are consistent and show that higher income households are most likely to experience food expenditure shocks in the presence of health shocks. This is due to households responding to health shocks by reallocating the households' funds away from food expenditure and towards the ill member (22). Our results do not show similar association within the lower income households (quintile 1) sample. This can be partially explained by the difference in healthcare access between income quintiles using the NIDS survey question *"where did the last (medical) consultation take place?"*. While 84% of responses within quintile 1 indicated public health facility, 56% of responses within quintile 2 indicated private health facility. The figures are even starker for the top 20% of households, where 86% indicated private health facility as the place of their last medical consultation. Of the quintile 2 households that accessed private healthcare, over 48% did not have medical insurance. This explains the

need for resource allocation away from food and the resulting food shock experienced among quintile 2 households.

This is therefore indicative that higher income households prefer to access private healthcare even at the cost of food consumption. However, the poor cannot use food expenditure as a coping mechanism to access private healthcare, as food consumption is already low. How the poor cope in South Africa in the event of a health shock in terms of non-medical expenses such as transport to public clinics or public hospitals needs to be studied further. The informal coping mechanisms as suggested by Udry [53], Townsend [54, 55] and Morduch [12] have not been explored in the South African context and provides avenues for future research. It appears from our research that the poor forego quality healthcare that requires additional expenditure because they are not able to employ diversion of food resources as it is already at a bare minimum. Conversely, higher income households show a definite preference for private healthcare and are prepared to incur additional expenditure even at the cost of food expenditure.

Furthermore, large-sized household and female headed households show greater vulnerability to food shocks. The presence of at least one employed household member also mitigates the health shock effect on food consumption. Overall, the study shows that the vulnerability of South African population to health shocks in the absence of universal medical insurance is precarious and costly to those most unable to afford private health insurance. Given that private healthcare is only available to a minority of the population, the vast majority of the populations is vulnerable because of a lack of healthcare insurance. This underlines and motivates the need for a National Health that would enable universal access to health care in the country.

## Supporting information

**S1 Table. Description of variables.**
(DOCX)

**S2 Table. Parallel trend assumption tests.**
(DOCX)

**S3 Table. PS-matching T-test results–logit.**
(DOCX)

## Author Contributions

**Conceptualization:** Umakrishnan Kollamparambil.

**Formal analysis:** Pheeha Morudu.

**Methodology:** Pheeha Morudu.

**Supervision:** Umakrishnan Kollamparambil.

**Writing – original draft:** Pheeha Morudu.

**Writing – review & editing:** Umakrishnan Kollamparambil.

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
