## [Decision Letter · Decision Letter 0]

30 Sep 2019

PONE-D-19-21685

Health Shocks, Medical Insurance and Household Vulnerability: Evidence from South Africa

PLOS ONE

Dear Dr. Kollamparambil,

Thank you for submitting your manuscript to PLOS ONE. After careful consideration, we feel that it has merit but does not fully meet PLOS ONE’s publication criteria as it currently stands. Therefore, we invite you to submit a revised version of the manuscript that addresses the points raised during the review process.

Thank you for considering PLOS One as the journal in which to publish your research.

Your manuscript has been reviewed favourably by two reviewers. The reviewers have raised specific issues regarding your manuscript that need to be resolved before the manuscript can be considered for publication in PLOS One. I would be grateful if you could consider the main issues regarding the layout and framing of this article and follow the PLOS One style as laid out in the Instructions for Authors. Reviewer 2 has provided some helpful suggestions for how these updates could be made.

Please respond to each of the reviewers' comments in turn taking particular note to address comments concerning: households' preference for public vs private healthcare; explain in more detail out-of-pocket expenditure and public-private mix in the context of the South African Health Care System and its funding; consider further explanation of use of reduced BMI in non-obese people as measure of health shock and also the pitfalls of food consumption estimates; and clearly elaborate on the difference-in-difference analysis performed (as per Reviewer 2's comments); and noting the important differences in the analyses by three different models.

Thanks again and we look forward to re-reviewing.

We would appreciate receiving your revised manuscript by Nov 14 2019 11:59PM. To enhance the reproducibility of your results, we recommend that if applicable you deposit your laboratory protocols in protocols.io, where a protocol can be assigned its own identifier (DOI) such that it can be cited independently in the future. For instructions see: http://journals.plos.org/plosone/s/submission-guidelines#loc-laboratory-protocols

We look forward to receiving your revised manuscript.

Kind regards,

Thomas Wingfield

Academic Editor

PLOS ONE

**Journal Requirements:**

**Comments to the Author**

1. Is the manuscript technically sound, and do the data support the conclusions?

Reviewer #1: Partly

Reviewer #2: Partly

2. Has the statistical analysis been performed appropriately and rigorously? 

Reviewer #1: Yes

Reviewer #2: No

3. Have the authors made all data underlying the findings in their manuscript fully available?

Reviewer #1: Yes

Reviewer #2: Yes

4. Is the manuscript presented in an intelligible fashion and written in standard English?

Reviewer #1: Yes

Reviewer #2: Yes

5. Review Comments to the Author

Reviewer #1: This paper analyses the relationship between health shocks in heads of households and household vulnerability in South Africa, using food expenditure as an indicator for vulnerability. The paper defined ‘health shock’ as a decline in BMI of more than one standard deviation in a non-obese head of household. The paper is well structured and easy to read.

Comments:

1) The authors assume that quality care can only be achieved in a private setting. An explanation for this is needed, as well as references.

2) On page 3, the authors state that 80% of South Africans are likely to be vulnerable to a sudden health shock. It is not clear how this percentage was calculated. Is this represented by the 100%- 16.9%= 83.1% of individuals not covered by medical aid in 2017? If yes, it should be made clearer.

3) BMI decline in non-obese individuals was defined as a health shock. However, this is a measure sensitive to other factors as well. BMI can also change as a result of ageing, change in lifestyle, seasonality (was the data collected during the same period of the year?), etc. It might be worth exploring this in a sensitivity analysis or recognising it as a limitation in the conclusion/discussion section.

4) Make sure all tables are referenced in text.

5) Last paragraph, page 11- it is not very clear if participants covered by medical aid experienced health shocks. If the answer is yes, it might be worth adding a few comments explaining possible reasons for this.

6) The authors comment throughout the paper that lower income households are not reducing their food consumption as a result of a health shock. However, there might be other explanations for this, such as consumption of basic food from own production, that is unaffected by illness. To account for this, it might be worth dividing food consumption into food purchased and food produced. Looking at the questionnaire for Wave 5 I think this is possible.

Reviewer #2: Reviewer: Nicola Foster

Manuscript number: PONE-D-19-21685

Title: Health shocks, medical insurance and household vulnerability: evidence from South Africa

Thank you for the opportunity to review the manuscript Health shocks, medical insurance and household vulnerability: evidence from South Africa.

It is really great to see the innovative ideas presented in this paper, and the questions identified are highly relevant for South Africa (and globally) for debates related to conceptualising Universal Health Coverage. The comments presented are aimed at strengthening the analysis presented here, and more broadly the conclusions drawn from the work. Comments are structured following the sections of the paper.

Overarching

• Currently, the manuscript reads as if written for an economics journal, and while this is not a problem, I would recommend that in order to make the paper more accessible to a wider audience, there is a need to use a style that is similar in structure to biomedical papers typically published in PLoS ONE. For each of the sections of the manuscript, I offer suggestions that may assist in improving this.

• In terms of the structure of the manuscript, perhaps you could consider amending headings to for example “measuring health shocks and household vulnerability” to “analytical framework”, “econometric strategy” perhaps rather as “methods”. Section 5. Data could possibly be subdivided to have a section “main results” and “robustness checks”. “6. Conclusion” could be renamed as “Discussion and Conclusion” as there is a decent proportion of section 6 that is in fact a detailed discussion of the results of the study.

Abstract

• Suggest rewriting the abstract to follow a format with introduction, methods, findings and conclusions. Currently, the paragraph opens with the objective of the study which feels out of place. It would improve the reading if you started with the global background to the study, for example the current debates around Universal Health Coverage (UHC).

• From the rest of the paper, I am not convinced that households are in fact showing a preference for private healthcare – is it possible that this is rather what they have access to? While all taxpayers (including through VAT) contribute to the financing of the public health sector in South Africa, those who are in formal employment are often enrolled into private health insurance by their employers.

• Please substantiate statement “the results confirm that access to quality healthcare is a privilege in South Africa”. It would be helpful if you could expand on how the quality of healthcare was conceptualised in this study and how implemented in this analysis?

Introduction

• Suggest adding a section that explicitly describes the characteristics of the South African health system. With an emphasis on the arrangement of and funding of health services between the public and private health system.

• If preferences for services (and how this intersects with affordability) is a thread that you would like to further explore in your work, then I suggest that this needs to be unpacked in more detail.

• It would be helpful to explicitly discuss out-of-pocket payments as one of the funding mechanisms for healthcare in South Africa – and how the burden of out of pocket payments affects people from different socio-economic status groups who are accessing health services from the public – and private health sectors. Perhaps you can also further expand on the funding for private health care services, which will include private health insurance, but also out of pocket payments? And perhaps a bit more on the range of benefit packages offered by various private health insurance schemes. For example, would households experience the same burden of out-of-pocket payment irrespective of the insurance scheme that they belong to? This would set up the introduction to your study better.

On page 3:

o In the figures related to medical aid coverage quoted, please specify enrolment in health insurance by type of insurance.

o Please provide a reference for the poverty rates quoted, and a definition for how poverty was defined here.

o Would be helpful to update the current developments related to the National Health Insurance in South Africa, especially the White Paper. Could you please reference the statement that the delays are related to sufficient public funding?

o Could you expand on your literature review related to vulnerability and food expenditure? In particular, could you unpack the direction of the relationships that have been explored in the literature you reviewed? One review to perhaps consider/ that may assist in identifying additional literature sources is Russell. 2004. The economic burden of illness for households in developing countries: a review of studies focusing on malaria, tuberculosis and HIV. In his review, he found that some of the studies relying on primary data collection, showed that household expenditure on food may increase when ill – but I think the question as to whether this may mean that there are intra-household shifts in food expenditure is less clear.

o The last paragraph before section 2, starting with “Given the binary nature of the dependent variable…”, would probably best fit in your main results section.

o On the definition of health shocks, I agree with the use of reductions in BMI as a measure, following the work of Wagstaff in Vietnam, however it would be helpful to expand a bit on your explanation of this measure. Could you say something about how relevant the work from Vietnam is likely to be to South Africa? Is this measure transferable and relevant, especially given the mixed evidence related to association and trend in South Africa between poverty and type II diabetes which is associated with higher BMIs? For example, Mutyambizi et al. 2019. Lifestyle and socio-economic inequalities in diabetes prevalence in South Africa: a decomposition analysis – find some evidence of an association between diabetes and higher SES in South Africa. However, Ataguba in an analysis of panel data, found a reduction in this trend over time. Is the focus on non-obese household heads, an attempt to control for the association with Diabetes, if so, please specify?

On page 6:

o Under econometric strategy, could you please explain here in which ways your study used a quasi-experimental data design?

On page 7:

o Given, the audience, it would be helpful to rewrite the sentence beginning with “Furthermore, given the panel data structure…” in simpler language. Especially, given that reverse causality could be used to wither mean that the direction of the cause and effect relationship is here the opposite of what one would normally consider, OR that there is a two-way causal relationship. I think that you mean the latter, but it would be helpful to be clear.

o With your difference-in-difference explanation, could you first succinctly explain what the first difference being assessed is; and then the second. An additional explanation would again assist readers without an economics background to understand your study better. Ideally this explanation should come before the equation (number 3) on page 7. Could you add a sentence to explain why you are using a non-linear DiD specifically with your question in mind?

On Page 10:

o Here, it would be worth reminding the reader that a health shock is defined as a significant reduction in BMI.

On Page 11:

o Table 2 would read better if you presented the comparison between Treated and Control in columns.

o Please be consistent in reporting the number of decimal points.

o Would suggest defining food shocks and how measured in the text here.

o In the table/ below the table, it would be helpful to repeat the definition of who the treated and control groups are again, i.e. those with a health shock vs those not.

Page 12:

o Remind the reader what a quintile one household is in the text. While it seems redundant, it really assists the readability.

o The reporting of changes between the groups could be improved by being consistent in reporting the decimal positions.

o Could you be specific about the threshold or the SA government pension grant (plus add a reference). How does that relate to the mean household income for each of your quintiles?

Page 13:

o In Table 3, again, it might be helpful to put the comparators (i.e. quintile 1 and quintile 2) on the horizontal axis; with the sample characteristics reported on the vertical axis.

o In your labelling, for each of the variables, it would be helpful if you could also specify the time period. So, for example, total medical expenditure per month? Per annum? In addition, please make sure that you report to the same number of decimal points.

o Page 13 (Table 3), could you please also discuss the finding that richer households has much greater total medical expenditure?

o Is “Head education attainment” a categorical variable? Or number of years of schooling? Please, clarify in the text.

o It would be helpful to also explain how medical aid coverage is defined i.e. does this include all patients who are accessing any form of health insurance whether it is full coverage or hospital only?

Tables 5, 6 and 7:

o For the reader, it would really improve the ability to see what is going on in the tables if you would add what the comparators are that are being reported.

Page 14:

o It would be helpful to substantiate from your results, why you conclude that “higher income households respond to health shocks by reallocating household spending towards the sick household member…”

o You may need to expand on how your results show that “higher income households have a preference to access better quality private sector healthcare”.

o Could your finding that poorer households do not have a reduction in food expenditure in the face of health shocks not also be explained by the protective effect of fee-free public health services in South Africa?

o Agree that this affect might be mediated by informal community support systems, and donations, but could it also be affected by labour behaviour. In the absence of out of pocket costs for healthcare, we would suspect that the largest cost contributor to health shocks would be loss of income. However, perhaps in South Africa (and especially in those informally employed) it might be that even significant reductions in BMI would not lead to a loss of income.

Page 19:

o The discussion/ conclusion section is well explained and points well argued.

o It would be helpful if you could add a section explaining the difference in the results between the three models used. Why was this important, what are the remaining limitations of the analysis and how does this affect your results?

6. PLOS authors have the option to publish the peer review history of their article (what does this mean?). If published, this will include your full peer review and any attached files.

Reviewer #1: Yes: Laura Rosu

Reviewer #2: Yes: Nicola Foster

---

## [Author Response · Author response to Decision Letter 0]

18 Nov 2019

The authors wish to thank the reviewers for their insightful comments and the opportunity to revise the manuscript. The authors have made every effort to comply with the comments received. Specific responses are provided below each comment.

Response TO:

Reviewer: Nicola Foster

Manuscript number: PONE-D-19-21685

Title: Health shocks, medical insurance and household vulnerability: evidence from South Africa

Thank you for the opportunity to review the manuscript Health shocks, medical insurance and household vulnerability: evidence from South Africa.

It is really great to see the innovative ideas presented in this paper, and the questions identified are highly relevant for South Africa (and globally) for debates related to conceptualising Universal Health Coverage. The comments presented are aimed at strengthening the analysis presented here, and more broadly the conclusions drawn from the work. Comments are structured following the sections of the paper. 

Overarching

• Currently, the manuscript reads as if written for an economics journal, and while this is not a problem, I would recommend that in order to make the paper more accessible to a wider audience, there is a need to use a style that is similar in structure to biomedical papers typically published in PLoS ONE. For each of the sections of the manuscript, I offer suggestions that may assist in improving this.

• In terms of the structure of the manuscript, perhaps you could consider amending headings to for example “measuring health shocks and household vulnerability” to “analytical framework”, “econometric strategy” perhaps rather as “methods”. Section 5. Data could possibly be subdivided to have a section “main results” and “robustness checks”. “6. Conclusion” could be renamed as “Discussion and Conclusion” as there is a decent proportion of section 6 that is in fact a detailed discussion of the results of the study. 

Changed headings in accordance to the comments. 

Abstract

• Suggest rewriting the abstract to follow a format with introduction, methods, findings and conclusions. Currently, the paragraph opens with the objective of the study which feels out of place. It would improve the reading if you started with the global background to the study, for example the current debates around Universal Health Coverage (UHC).

Restructured the abstract accordingly.

From the rest of the paper, I am not convinced that households are in fact showing a preference for private healthcare – is it possible that this is rather what they have access to? While all taxpayers (including through VAT) contribute to the financing of the public health sector in South Africa, those who are in formal employment are often enrolled into private health insurance by their employers. 

Public healthcare is freely available all residents in South Africa. Those with private health insurance have the option of choosing public health facility over private facility with no additional cost implication. The NIDS data however indicates that only 4% of private insurance holders consulted public health facility (this is explained by the limited nature of some insurance packages ). On the contrary, over 22% of non-insurance holders were willing to incur costs to access private facility. This can be construed as a preference for private health facility over public health facility. 

• Please substantiate statement “the results confirm that access to quality healthcare is a privilege in South Africa”. It would be helpful if you could expand on how the quality of healthcare was conceptualised in this study and how implemented in this analysis? 

References have been added that indicate higher satisfaction level for private health users as compared to public health users. 

Introduction

• Suggest adding a section that explicitly describes the characteristics of the South African health system. With an emphasis on the arrangement of and funding of health services between the public and private health system.

Added

• If preferences for services (and how this intersects with affordability) is a thread that you would like to further explore in your work, then I suggest that this needs to be unpacked in more detail

Added

• It would be helpful to explicitly discuss out-of-pocket payments as one of the funding mechanisms for healthcare in South Africa – and how the burden of out of pocket payments affects people from different socio-economic status groups who are accessing health services from the public – and private health sectors. Perhaps you can also further expand on the funding for private health care services, which will include private health insurance, but also out of pocket payments? And perhaps a bit more on the range of benefit packages offered by various private health insurance schemes. For example, would households experience the same burden of out-of-pocket payment irrespective of the insurance scheme that they belong to? This would set up the introduction to your study better.

o This is an important matter as all insurance schemes do not offer the same coverage. The NIDS data however do not offer the details on medical insurance coverage and this is cited in the limitation of the study.

On page 3:

o In the figures related to medical aid coverage quoted, please specify enrolment in health insurance by type of insurance.

o As indicated above, the NIDS data does not give details on the type of insurance 

o Please provide a reference for the poverty rates quoted, and a definition for how poverty was defined here.

Added.

o Would be helpful to update the current developments related to the National Health Insurance in South Africa, especially the White Paper. Could you please reference the statement that the delays are related to sufficient public funding?

Referenced and added some updates.

o Could you expand on your literature review related to vulnerability and food expenditure? In particular, could you unpack the direction of the relationships that have been explored in the literature you reviewed? One review to perhaps consider/ that may assist in identifying additional literature sources is Russell. 2004. The economic burden of illness for households in developing countries: a review of studies focusing on malaria, tuberculosis and HIV. In his review, he found that some of the studies relying on primary data collection, showed that household expenditure on food may increase when ill – but I think the question as to whether this may mean that there are intra-household shifts in food expenditure is less clear. 

Included discussion on increased food expenditure due to “special food” expenditure to assist with quick recovery of patent. Also introduced discussions on “medical poverty trap”.

o The last paragraph before section 2, starting with “Given the binary nature of the dependent variable…”, would probably best fit in your main results section.

Moved accordingly.

o On the definition of health shocks, I agree with the use of reductions in BMI as a measure, following the work of Wagstaff in Vietnam, however it would be helpful to expand a bit on your explanation of this measure. Could you say something about how relevant the work from Vietnam is likely to be to South Africa? Is this measure transferable and relevant, especially given the mixed evidence related to association and trend in South Africa between poverty and type II diabetes which is associated with higher BMIs? For example, Mutyambizi et al. 2019. Lifestyle and socio-economic inequalities in diabetes prevalence in South Africa: a decomposition analysis – find some evidence of an association between diabetes and higher SES in South Africa. However, Ataguba in an analysis of panel data, found a reduction in this trend over time. Is the focus on non-obese household heads, an attempt to control for the association with Diabetes, if so, please specify?

By excluding the obese individuals we have been able to account for the high proportion of obesity in South Africa resulting from various factors including diabetes and also unhealthy diet. The association of both diabetes and food habits with SES is debatable. This is further explained in the study for clarity.

On page 6:

o Under econometric strategy, could you please explain here in which ways your study used a quasi-experimental data design? 

Explained under Methods section.

On page 7:

o Given, the audience, it would be helpful to rewrite the sentence beginning with “Furthermore, given the panel data structure…” in simpler language. Especially, given that reverse causality could be used to wither mean that the direction of the cause and effect relationship is here the opposite of what one would normally consider, OR that there is a two-way causal relationship. I think that you mean the latter, but it would be helpful to be clear.

Attempted to rephrase.

o With your difference-in-difference explanation, could you first succinctly explain what the first difference being assessed is; and then the second. An additional explanation would again assist readers without an economics background to understand your study better. Ideally this explanation should come before the equation (number 3) on page 7. Could you add a sentence to explain why you are using a non-linear DiD specifically with your question in mind?

Explained when introducing DID and added the sentence. 

On Page 10:

o Here, it would be worth reminding the reader that a health shock is defined as a significant reduction in BMI.

Reminded the reader what health shock is.

On Page 11:

o Table 2 would read better if you presented the comparison between Treated and Control in columns.

Done

o Please be consistent in reporting the number of decimal points.

The within reporting are kept them at 4 decimals, however, for money values it made sense to keep them at 2 decimals. However, in body decimals are restricted to 2 decimals. 

o Would suggest defining food shocks and how measured in the text here.

Done

o In the table/ below the table, it would be helpful to repeat the definition of who the treated and control groups are again, i.e. those with a health shock vs those not.

Defined it before the table.

Page 12:

o Remind the reader what a quintile one household is in the text. While it seems redundant, it really assists the readability.

Included the explanations

o The reporting of changes between the groups could be improved by being consistent in reporting the decimal positions. 

Within table is kept at 4 decimal places except for monetary values.

o Could you be specific about the threshold or the SA government pension grant (plus add a reference). How does that relate to the mean household income for each of your quintiles?

Added reference and related it to the household income. 

Page 13:

o In Table 3, again, it might be helpful to put the comparators (i.e. quintile 1 and quintile 2) on the horizontal axis; with the sample characteristics reported on the vertical axis.

The comparators are on the horizontal. 

o In your labelling, for each of the variables, it would be helpful if you could also specify the time period. So, for example, total medical expenditure per month? Per annum? In addition, please make sure that you report to the same number of decimal points.

The expenditure values are for the “last 30 days” of the survey period. Mentioned that in text. 

o Page 13 (Table 3), could you please also discuss the finding that richer households has much greater total medical expenditure?

Included possible explanation for this.

o Is “Head education attainment” a categorical variable? Or number of years of schooling? Please, clarify in the text.

In regression analysis education attainment is a Dummy variable for each level (no education, primary, secnoday, matriculation, technical, university). In summary tables “Head educational attainment” is in years of education (cardinal variable). Details Included in variable definition in appendix.

o It would be helpful to also explain how medical aid coverage is defined i.e. does this include all patients who are accessing any form of health insurance whether it is full coverage or hospital only?

NIDS data (manual and surveys) does not clarify the kind of coverage this is. Whether it partial or full coverage. This is a limitation and can explain why a small proportion of medical insurance holders access public health facilities.

Tables 5, 6 and 7:

o For the reader, it would really improve the ability to see what is going on in the tables if you would add what the comparators are that are being reported.

Now Table 4,5,6. The comparators are added, and to try to make it clearer, titles have been adjusted to show which models each table corresponds to. And removed the numbering within tables. 

Page 14:

o It would be helpful to substantiate from your results, why you conclude that “higher income households respond to health shocks by reallocating household spending towards the sick household member…”

cited additional literature to explain this 

o You may need to expand on how your results show that “higher income households have a preference to access better quality private sector healthcare”. 

o Could your finding that poorer households do not have a reduction in food expenditure in the face of health shocks not also be explained by the protective effect of fee-free public health services in South Africa?

Yes, public health facilities are free in South Africa to all residents irrespective of Income criteria. However 14% of quintile 1 individuals without medical insurance seek private healthcare, 37% percent of quintile 2 individuals with medical insurance are accessing healthcare in private facility. In other words, 48% of quintile 2 individuals who access private health facilities, do not have medical insurance. This is indicative of preference for private over public healthcare.

 medical aid public private other

 No % 84 13.93 2

q1 Yes % 43 56.76 0

 All % 84 14.64 2

 No % 62 37.28 1

q2 Yes % 7 92.39 1

 All % 45 54.23 1

o Agree that this affect might be mediated by informal community support systems, and donations, but could it also be affected by labour behaviour. In the absence of out of pocket costs for healthcare, we would suspect that the largest cost contributor to health shocks would be loss of income. However, perhaps in South Africa (and especially in those informally employed) it might be that even significant reductions in BMI would not lead to a loss of income.

It is possible, but not certain, that health shock leads to a loss of income which in turn could result in food shock. This was controlled in the covariates using the dummy variable for employed individuals. 

Page 19:

o The discussion/ conclusion section is well explained and points well argued.

Response TO:

Reviewer #1: Laura Rosu

Reviewer #1: This paper analyses the relationship between health shocks in heads of households and household vulnerability in South Africa, using food expenditure as an indicator for vulnerability. The paper defined ‘health shock’ as a decline in BMI of more than one standard deviation in a non-obese head of household. The paper is well structured and easy to read.

Comments:

1) The authors assume that quality care can only be achieved in a private setting. An explanation for this is needed, as well as references.

Included the following: The Parliamentary Monitoring Group in 2016 reported after an inspection of 10% of public health faclites that “Common to all facilities were challenges regarding patient safety being compromised, good pharmacy practice (GPP) not being adhered to, waste mismanagement, lack of cleanliness, as well as poor maintenance of grounds and equipment” (1). Other studies comparing public and private facilities (2) highlight the major gap between private and public healthcare in South Africa. Public healthcare has many disadvantages such as long wait times, poor quality of care, rushed apppointments, old facilities, and poor disease control and prevention practices. Private healthcare on the other hand is expensive, but has short wait times, quality care, better facilities, adequate resources available, appointments are not rushed, and proper disease control and prevention practices are utilized. Another study among older South Africans

 (3) indicates that healthcare responsiveness perception was higher in private than in public inpatient and outpatient healthcare facilities.

1) PMG (2016) Public Health Facilities audit results: Office of Health Standards Compliance (OHSC) briefing, Parliamentary Monitoring Group, South Africa. https://pmg.org.za/committee-meeting/22233/

2) Young M ( 2016) Private vs. Public Healthcare in South Africa, Honours Theses Paper 2741, Western Michigan University. https://scholarworks.wmich.edu/cgi/viewcontent.cgi?article=3752&context=honors_theses

3) Karl Peltzer & Nancy Phaswana-Mafuya (2012) Patient experiences and health system responsiveness among older adults in South Africa, Global Health Action, 5:1, DOI: 10.3402/gha.v5i0.18545 

2) On page 3, the authors state that 80% of South Africans are likely to be vulnerable to a sudden health shock. It is not clear how this percentage was calculated. Is this represented by the 100%- 16.9%= 83.1% of individuals not covered by medical aid in 2017? If yes, it should be made clearer.

Yes, made clearer.

3) BMI decline in non-obese individuals was defined as a health shock. However, this is a measure sensitive to other factors as well. BMI can also change as a result of ageing, change in lifestyle, seasonality (was the data collected during the same period of the year?), etc. It might be worth exploring this in a sensitivity analysis or recognising it as a limitation in the conclusion/discussion section.

Included it as a limitation in the discussion section.

4) Make sure all tables are referenced in text.

Tables are referenced in text.

5) Last paragraph, page 11- it is not very clear if participants covered by medical aid experienced health shocks. If the answer is yes, it might be worth adding a few comments explaining possible reasons for this.

health 

shock medical aid dummy 

dummy 0 (%) 1 (%) Total

0 1,305 455 1,760

 (97.75) (97.85) 97.78

1 30 10 40

 (2.25) (2.15) 2.22

Total 1,335 465 1,800

 (100) (100) 100

Included the following: There is no significant difference in the proportion of sample that experienced health shocks by their medical insurance status. Evidence in literature is mixed in terms of the effect of medical insurance on health status, with a recent review (1) indicating that of the 12 studies: nine studies found a positive effect, one study reported a negative effect, and two studies reported no effect. Therefore the indication that medical insurance does not make a substantial impact on health shock is not contradictory from existing evidence.

(1) Erlangga D, Suhrcke M, Bloor K, Ali S (2019) The impact of public health insurance on health care utilisation, financial protection and health status in low- and middle-income countries: A systematic review. PLoS ONE 14(8): e0219731. https://doi.org/10.1371/journal.pone.0219731

6) The authors comment throughout the paper that lower income households are not reducing their food consumption as a result of a health shock. However, there might be other explanations for this, such as consumption of basic food from own production, that is unaffected by illness. To account for this, it might be worth dividing food consumption into food purchased and food produced. Looking at the questionnaire for Wave 5 I think this is possible.

We introduced an additional binary variable among the covariates to control for food production by the household. The coefficient of the new variable is insignificant and does not change the results of our key variables. This is not surprising because just over 5% of the sample reported some element of home production (either agricultural or animal related produce) of food. The results are reported in table below but we prefer to use the original model keeping in mind the principle of parsimony in model specification.

 (All sample) (Quintile 1) (Quintile 2)

VARIABLES food shock food shock food shock

postperiod 0.449** 0.655*** -0.0234

 (0.180) (0.198) (0.365)

treatmentA -1.552*** -1.117* -2.714**

 (0.550) (0.598) (1.249)

treatpost 2.345*** 1.348 5.786***

 (0.861) (0.959) (1.713)

w5_h_fdprd -0.0272 0.349 -0.489

(food production) (0.215) (0.406) (0.631)

medicaid -0.402 -1.851* 0.155

 (0.320) (1.021) (0.424)

female -0.352** -0.638*** 0.113

 (0.179) (0.200) (0.308)

married 0.124 0.382 -0.787

 (0.234) (0.259) (0.503)

ageb 0.0569 0.137 -0.281

 (0.339) (0.452) (0.490)

agec -0.190 -0.237 -0.172

 (0.385) (0.495) (0.621)

aged 0.439 0.126 1.368

 (0.582) (0.587) (0.938)

educb 0.464* 0.481* 0.118

 (0.258) (0.274) (0.868)

educc -0.0891 -0.124 -0.315

 (0.309) (0.349) (0.767)

educd -0.212 0.141 -0.576

 (0.414) (0.472) (0.898)

educe 0.483 1.091 0.300

 (0.426) (0.751) (0.813)

educf -1.176 -1.406

 (0.836) (1.151)

hhsb 0.689*** 0.391* 0.810*

 (0.191) (0.221) (0.424)

hhsc 0.608** 0.278 1.015*

 (0.240) (0.259) (0.614)

hhsd 0.886*** 0.518 2.853**

 (0.316) (0.343) (1.124)

pensionrecp -0.696* -0.462 -0.795

 (0.377) (0.317) (0.825)

employment -0.296 -0.232 0.894*

 (0.203) (0.231) (0.505)

urban 0.234 0.271 0.311

 (0.205) (0.265) (0.352)

toileth -0.171 -0.186 0.101

 (0.216) (0.270) (0.411)

waterp 0.0885 0.123 -0.0936

 (0.265) (0.270) (0.608)

Constant -2.405*** -2.793*** -2.779

 (0.586) (0.953) (1.777)

Observations 4,771 3,067 1,701

---

## [Decision Letter · Decision Letter 1]

7 Jan 2020

Health Shocks, Medical Insurance and Household Vulnerability: Evidence from South Africa

PONE-D-19-21685R1

Dear Dr. Kollamparambil,

We are pleased to inform you that your manuscript has been judged scientifically suitable for publication and will be formally accepted for publication once it complies with all outstanding technical requirements.

With kind regards,

Thomas Wingfield

Academic Editor

PLOS ONE

Additional Editor Comments (optional):

The reviewers and I have accepted your manuscript for publication in PLOS One and would like to take the opportunity to congratulate you on your important work. The only comment by one of the reviewers that should be addressed during the production stages is: "Please provide a reference for the statement that private health care in South Africa is of good quality. "Private healthcare on the other hand is expensive, but provide quality facilities and care." Quality is a highly subjective description. Given the measuring yard used for public health facilities in that same paragraph, and the statement is contrasting the two - has a similar (independent) audit been conducted on private health care facilities? If so, it would provide balance to that assessment if such work could be quoted and referenced in the text. Alternatively, would suggest rephrasing the statement.

We look forward to seeing your research in print.

Yours sincerely,

Tom Wingfield

Reviewers' comments:

Reviewer's Responses to Questions

**Comments to the Author**

1. If the authors have adequately addressed your comments raised in a previous round of review and you feel that this manuscript is now acceptable for publication, you may indicate that here to bypass the “Comments to the Author” section, enter your conflict of interest statement in the “Confidential to Editor” section, and submit your "Accept" recommendation.

Reviewer #1: All comments have been addressed

Reviewer #2: All comments have been addressed

2. Is the manuscript technically sound, and do the data support the conclusions?

Reviewer #1: Yes

Reviewer #2: Partly

3. Has the statistical analysis been performed appropriately and rigorously? 

Reviewer #1: Yes

Reviewer #2: N/A

4. Have the authors made all data underlying the findings in their manuscript fully available?

Reviewer #1: Yes

Reviewer #2: Yes

5. Is the manuscript presented in an intelligible fashion and written in standard English?

Reviewer #1: Yes

Reviewer #2: Yes

6. Review Comments to the Author

Reviewer #1: The authors have nicely addressed all comments, so I have no further comments.

Reviewer #2: Please provide a reference for the statement that private health care in South Africa is of good quality. "Private healthcare on the other hand is expensive, but provide quality facilities and care."

Quality is a highly subjective description. Given the measuring yard used for public health facilities in that same paragraph, and the statement is contrasting the two - has a similar (independent) audit been conducted on private health care facilities? If so, it would provide balance to that assessment if such work could be quoted and referenced in the text. Alternatively, would suggest rephrasing the statement.

7. PLOS authors have the option to publish the peer review history of their article (what does this mean?). If published, this will include your full peer review and any attached files.

Reviewer #1: Yes: Laura Rosu

Reviewer #2: Yes: Nicola Foster

---

## [Editor Report · Acceptance letter]

21 Jan 2020

PONE-D-19-21685R1 

Health Shocks, Medical Insurance and Household Vulnerability: Evidence from South Africa 

Dear Dr. Kollamparambil:

I am pleased to inform you that your manuscript has been deemed suitable for publication in PLOS ONE. Congratulations! Your manuscript is now with our production department. 

With kind regards,

on behalf of

Dr. Thomas Wingfield 

Academic Editor

PLOS ONE